# LRRC8A Is a Promising Prognostic Biomarker and Therapeutic Target for Pancreatic Adenocarcinoma

**DOI:** 10.3390/cancers14225526

**Published:** 2022-11-10

**Authors:** Rong Xu, Yaohua Hu, Qinghua Xie, Caiqin Zhang, Yong Zhao, He Zhang, Hailong Shi, Xiaoming Wang, Changhong Shi

**Affiliations:** 1Division of Cancer Biology, Laboratory Animal Center, The Fourth Military Medical University, Xi’an 710032, China; 2Department of Geriatrics, Xijing Hospital, The Fourth Military Medical University, Xi’an 710032, China; 3School of Basic Medical Sciences, Medical College of Yan’an University, Yan’an 716000, China; 4Laboratory Animal Center, Guangzhou University of Chinese Medicine, Guangzhou 510405, China; 5Laboratory Animal Center, General Hospital of Northern Theater Command, Shenyang 110016, China; 6College of Basic Medicine, Shaanxi University of Chinese Medicine, Xianyang 712046, China

**Keywords:** pancreatic adenocarcinoma (PAAD), bioinformatics analyses, leucine-rich repeat-containing protein 8A (LRRC8A), immune infiltration

## Abstract

**Simple Summary:**

Pancreatic adenocarcinoma (PAAD) is a highly malignant tumor with a poor prognosis and lack of effective biomarkers. We demonstrated, using bioinformatics analyses, that the leucine-rich repeat-containing 8A (LRRC8A) can affect the prognosis of patients with PAAD. In addition, experiments in vitro showed shRNA of LRRC8A can significantly affect the hallmarks of PAAD, including cell proliferation, cell migration, drug resistance, and immune infiltration. Collectively, our data suggest that LRRC8A could be a promising prognostic biomarker and therapeutic target for PAAD.

**Abstract:**

Pancreatic adenocarcinoma (PAAD) is a highly malignant tumor of the digestive system with increasing morbidity and mortality. The lack of sensitive and reliable biomarkers is one of the main reasons for the poor prognosis. Volume-regulated anion channels (VRAC), which are ubiquitously expressed in the vertebrate cell membrane, are composed of leucine-rich repeat-containing 8A (LRRC8A) and four other homologous family members (LRRC8B–E). VRAC heterogeneous complex is implicated in each of the six “hallmarks of cancer” and represents a novel therapeutic target for cancer. In this study, LRRC8A was speculated to be a promising prognostic biomarker and therapeutic target for PAAD based on a series of bioinformatics analyses. Additional cell experiments and immunohistochemical assays demonstrated that LRRC8A can affect the prognosis of PAAD and is correlated to cell proliferation, cell migration, drug resistance, and immune infiltration. Functional analysis indicated that LRRC8A influences the progression and prognosis of patients with PAAD by the regulation of CD8^+^ T cells immune infiltration. Taken together, these results can help in the design of new therapeutic drugs for patients with PAAD.

## 1. Introduction

Pancreatic adenocarcinoma (PAAD) is a deadly malignant tumor, with a 5-year survival rate of only 2–9%. It is the fourth leading cause of cancer-associated mortality [1,2]. Most patients with PAAD remain asymptomatic during the initial stage of cancer development until they progress to advanced metastasis, wherein tumor cells become highly invasive [3]. The lack of sensitive and reliable biomarkers is one of the main reasons for its poor prognosis. Although the application of carbohydrate antigen (CA) 19-9, which is the primary biomarker used at present for PAAD has witnessed incremental progress in the diagnosis of PAAD, it has a limitation that cannot be overlooked, wherein CA19-9 showed false-positive results in patients with chronic pancreatitis, cholangitis, and other diseases [4]. Several novel biomarkers, including HOX antisense intergenic RNA (a type of long non-coding RNA), C4b binding protein alpha, micro-RNA, and exosomes have been identified as potential PAAD biomarkers, but these markers cannot guarantee high reliability, sensitivity, and specificity owing to some inherent limitations, such as inadequate quantities of micro RNA in various biofluids [5,6,7]. Another challenge in the treatment of PAAD is the complex tumor microenvironment, including the presence of tumor-associated macrophages and regulatory T cells within the stroma that create a barrier for conventional therapies and immunotherapies [8]. Although numerous clinical trials targeting the stroma have been conducted, most of them did not provide the expected efficacy based on preclinical studies [9,10]. Therefore, identifying novel biomarkers to predict the prognosis of PAAD and developing effective therapeutic targets are required for the treatment of PAAD.

Volume-regulated anion channels (VRACs), also called volume-sensitive outwardly rectifying anion channels, are ubiquitously expressed in the membrane of vertebrate cells. Swelling-activated chloride currents are activated when VRACs are opened. Leucine-rich repeat-containing 8A (LRRC8A) and four other homologous family members (LRRC8B–E) assemble into VRAC complexes. Since these proteins can exist in a soluble form and as an integral membrane protein, cyst fluid sampling and testing for LRRC8A and its homologous family proteins are feasible using mass spectroscopy. This would provide a potential opportunity for early detection and management [11]. Moreover, as the main chloride channel type, VRAC is implicated in each of the six “hallmarks of cancer” and represents an emerging therapeutic target for cancer [12,13,14]. A dysregulated expression or dysfunction of VRAC can be detected in many cancer types, indicating a correlation between ion channels and tumorigenesis [15]. LRRC8A downregulation can reportedly increase the sensitivity of glioblastoma to temozolomide and carmustine [16]. LRRC8A can regulate the proliferation and metastasis of colorectal cancer, hepatocellular carcinoma, and nasopharyngeal carcinoma [16,17,18,19]. Recent studies also supported the finding that LRRC8D and LRRC8A/LRRC8E play an integral role in the regulation of multidrug resistance and tumor immunotherapy, respectively [20]. Previous studies confirmed the role of calcium-activated chloride channel regulator 1 as a potential suppressor and independent factor in PAAD [21]; however, there are no data regarding the prognostic role of VRAC in PAAD.

In this study, a series of bioinformatics analyses based on several large public databases were conducted, and LRRC8A was identified as a possible potent therapeutic target and prognostic biomarker for PAAD based on cell experiments. Additional functional analysis, immune infiltration analysis, and immunohistochemical assay indicated that LRRC8A influences the progression and prognosis of PAAD by the regulation of CD8^+^ T cells immune infiltration.

## 2. Materials and Methods

### 2.1. Pan-Cancer Analysis of LRRC8A-8E Expression by ONCOMINE

The mRNA expression of LRRC8A, LRRC8B, LRRC8C, LRRC8D, and LRRC8E in different human cancer types was analyzed using ONCOMINE (www.oncomine.org (accessed on 11 June 2021)). A *p*-value of 0.05 and fold change of 2 were set as significance thresholds.

### 2.2. LRRC8A–8E Expression in PAAD by GEPIA and UALCAN

The transcriptional levels of LRRC8A–8E in PAAD were thoroughly and comprehensively analyzed using GEPIA (http://gepia.cancer-pku.cn/ (accessed on 27 July 2021)) and UALCAN (http://ualcan.path.uab.edu/analysis.html (accessed on 27 July 2021)). The expression data of tumor samples (n = 179) and normal samples (n = 171) in GEPIA were acquired from the Cancer Genome Atlas (TCGA) and the Genotype-Tissue Expression. Each box plot of LRRC8A–8E in PAAD was designed using the “Expression DIY” module of GEPIA. The expression levels of LRRC8A–8E based on individual cancer stages in PAAD were analyzed using the TCGA dataset in UALCAN. *p* < 0.05 was set as a significance threshold.

### 2.3. Survival Analysis Using Kaplan–Meier Plotter and GEPIA

The correlation between LRRC8A–8E expression and clinical outcomes of PAAD was assessed using a Kaplan–Meier Plotter (https://kmplot.com/analysis/ (accessed on 7 January 2022)) and the module of survival analysis in GEPIA. The Kaplan–Meier plotter can assess the overall survival (OS) and relapse-free survival (RFS) of patients with PAAD (n = 177), whereas the OS and disease-free survival (DFS) analysis of PAAD can be obtained using GEPIA (n = 178).

### 2.4. Functional Analysis of LRRC8A–8E in Patients with PAAD

The co-expression and network module of LRRC8A–E in patients with PAAD were analyzed using cBioPortal (www.cbioportal.org (accessed on 27 July 2021)), which included 11 studies with 1347 PAAD samples. The Gene Ontology (GO) enrichment heatmap and the Kyoto Encyclopedia of Genes and Genomes (KEGG) pathway analysis of the top 50 genes positively associated with LRRC8A were performed using OmicShare tools, which is a free online platform for data analysis (https://www.omicshare.com (accessed on 25 April 2022)). The protein–protein interaction (PP—network of LRRC8A–E was analyzed using the Search Tool for Retrieval of Interacting Genes/Proteins (https://string-db.org/ (accessed on 23 July 2021)) and GeneMANIA (http://www.genemania.org (accessed on 27 July 2021)).

### 2.5. Correlations between LRRC8A–E Expression and Immune Infiltration Using Tumor Immune Estimation Resource (TIMER)

GEPIA databases and TIMER (http://cistrome.org/TIMER/ (accessed on 28 July 2021)) were used to determine the relationship between LRRC8A–E expression and immune infiltration. The TIMER database contains 10,897 samples across 32 cancer types from TCGA to allow the evaluation of the abundance of immune infiltration.

### 2.6. Cell Lines

PAAD cell lines PANC1 (CRL-1469™), AsPC-1 (CRL-1682™), and BxPC-3 (CRL-1687) were obtained from the American Type Culture Collection (USA), AsPC-1^GEM-R^ cell line (gemcitabine-resistant AsPC-1) was established in our laboratory, which can grow in the medium containing 18 μg/mL gemcitabine for more than 4 days. All cell lines were cultured in Dulbecco’s modified Eagle’s medium supplemented with 10% fetal bovine serum. Mycoplasma was detected and determined to be negative.

### 2.7. Immunochemistry (IHC) Analysis

PAAD tissues were obtained from the Department of Pathology of Xijing Hospital, and patient consent was reviewed and approved by the Xijing Hospital Clinical Research Ethics Board (certificate number KY2015432). Based on standard protocols [22], IHC analysis was performed using antibodies such as LRRC8A (Biobyt, Cambridge, United Kingdom; orb185053) and CD8 (Proteintech, Rosemont, IL, USA; 66868-1-Ig). The area of positive cells, average optical density values, and integrated optical density values were calculated using Image-Pro Plus analysis software 2020 to evaluate the immunohistochemical results. The R software (version 4.1.0) was used to linearly fit the immunohistochemical quantification data and the function Cook’s distance was used for data visualization.

### 2.8. Real-Time Polymerase Chain Reaction (PCR) Assays

Real-time PCR was performed using a PrimeScript™ RT Reagent Kit and TB Green^®^ Fast qPCR Mix purchased from Takara (Beijing, China) using a Step One Plus real-time PCR Detection System (Applied Biosystems, Thermo Fisher, Waltham, MA, USA). Relative mRNA expression levels were calculated using the 2^−ΔΔCT^ method, and β-actin mRNA expression was set as the internal reference. The sequences of primers used for PCR are shown in Table 1.

### 2.9. Western Blotting Analysis

Western blotting analyses were performed using standard protocols [23]. Antibodies against the following proteins were used: LRRC8A (Biobyt, orb185053) and β-actin (Engibody, Dover, DE, USA; AT0001). A grayscale value of western blotting was obtained using the Image J software 2020 and analyzed using the GraphPad Prism 6 software.

### 2.10. LRRC8A Knockdown

The knockdown of LRRC8A was accomplished by employing short hairpin (shRNA) lentivirus (sense: CCGUCUACUACGUGCACAATT, antisense: UUGUGCACGUAGUAGACGGTT) (Hanbio, Shanghai, China) infection as reported previously [24], and this was verified by real-time PCR and western blotting analysis.

### 2.11. Cell Viability Assay

Cells were seeded in a 96-well plate and pretreated with 10 μM 4-(2-Butyl-6, 7-dichloro-2-cyclopentyl-indan-1-on-5-yl) oxobutyric acid (DCPIB), a kind of specific VRAC inhibitor. Then, the cells were treated with 100 μM gemcitabine, and cell viability was detected at different times using a Cell Counting Kit-8 (CCK-8) (Solarbio, Beijing, China; CA1210).

### 2.12. Wound Healing Assay

AsPC-1 and BxPC-3 cells were seeded in a six-well plate and infected with ShRNA lentivirus of LRRC8A. When the cells reached approximately 80% confluence, a pipette tip was used to create a straight line in the middle of the plate. Then, the shed cells were rinsed using phosphate-buffered saline, and a serum-free medium was added. Images of the straight line at 0 h and 48 h were captured, and the relative migration distance was analyzed to elucidate the scribing process of the cells. AsPC-1 and BxPC-3 cells infected with scramble shRNA lentivirus were used as the negative control.

### 2.13. Transwell Assay

AsPC-1 and BxPC-3 cells with knocked-down LRRC8A were added to the chamber and cultured for 48 h. Then, the cells were treated with 4% paraformaldehyde and stained with crystal violet. Images of cells that passed through the chamber were captured under a microscope.

### 2.14. Statistical Analysis

All data are presented as means ± standard error of the mean and analyzed using the GraphPad Prism 6 software. The significance was analyzed using Student’s *t*-tests. *p* < 0.05 was considered statistically significant.

## 3. Results

### 3.1. LRRC8A Is Highly Expressed in PAAD in Pan-Cancer Analysis

The expression of LRRC8A and four other homologous family members (LRRC8B–E) in pan-cancer was retrieved using the ONCOMINE database (Figure 1a). Results revealed that LRRC8A expression was higher in PAAD samples than in normal samples. To further evaluate its expression in PAAD, GEPIA, and UALCAN were used. As shown in Figure 1b, the transcripts per million expression level of LRRC8A in PAAD samples was seven folds higher than that in normal samples (46.52 in tumor samples vs. 6.59 in normal samples). Subsequently, an in-depth analysis by UALCAN demonstrated a significant correlation between LRRC8A expression (*p ** < 0.05) and the pathological stage of patients with PAAD (Figure 1c). The expression of LRRC8B–8E in different pathological stages of PAAD is shown in Appendix A.

### 3.2. LRRC8A Expression Is Associated with Poor Prognosis in Patients with PAAD

To evaluate the value of differentially expressed LRRC8A–E in the progression of PAAD, its correlation with clinical outcomes was assessed using GEPIA (n = 178) and Kaplan–Meier plotter (n = 177). Using GEPIA, the DFS (LOGRANK test, *p* = 0.0052; hazards ratio (HR), 1.9 (high); *p* (HR) = 0.006) and OS rates (logrank test, *p* = 0.017; HR, 1.7 (high); *p* (HR) = 0.018) of patients with PAAD with a high LRRC8A expression were significantly worse than those with a low LRRC8A expression (Figure 1d,e). Similarly, the RFS of patients with PAAD using a Kaplan–Meier plotter also indicated that LRRC8A was a detrimental prognostic factor of PAAD (RFS: HR, 4.13, 95% CI, 1.6–10.69; logrank *p* = 0.0016) (Figure 1f). The correlation of LRRC8B–8E with clinical outcomes of PAAD is shown in Appendix A.

### 3.3. Identification of the Role of LRRC8A in PAAD In Vitro

As mentioned above, uncontrolled proliferation, frequent migration, and multidrug resistance are the fundamental hallmarks of cancer cells. A gemcitabine-resistant AsPC-1^GEMR^ cell line was established to investigate whether LRRC8A was involved in the drug resistance of PAAD (Figure 2a). The CCK-8 assay results revealed that the knockdown of LRRC8A using shRNA or DCPIB, the specific VRAC inhibitor can inhibit the proliferation of PAAD cells and increase their sensitivity to gemcitabine (Figure 2b, Appendix A). Additionally, qRT-PCR and western blot results showed LRRC8A decreased after AsPC-1^GEMR^ cells were pretreated with DCPIB (Appendix A) and the knockdown effect of shLRRC8A in PAAD cell lines was showed in Appendix A.

To evaluate the potential role of LRRC8A in PAAD, qRT-PCR and western blotting were used to detect the expression of LRRC8A in PAAD cell lines AsPC-1, PaNC-1, and BxPC-3. The sequence of cell migration and invasion capacity, angiogenic potential, and tumorigenicity was reportedly BxPC-3 > AsPC-1 > PaNC-1 [25]. A similar mRNA and protein expression gradient of LRRC8A in PAAD cell lines was also observed in Figure 2c,d, respectively. The immunohistochemical analysis of pathological sections obtained from patients with PAAD demonstrated that a high LRRC8A expression was correlated to the tumor grade of PAAD (Figure 2e). To investigate whether LRRC8A affects the migration and invasion abilities of PAAD cells, a wound healing assay with or without LRRC8A knockdown was conducted using AsPC-1^GEMR^ and BxPC-3 cells as showed in Figure 2f, and a transwell invasion assay using AsPC-1 and BxPC-3 cells with or without LRRC8A knockdown were performed in Figure 2g. Results showed a substantial inhibitory effect of LRRC8A knockdown on PAAD cell migration and invasion (Figure 2f,g). Further investigation showed LRRC8A knockdown resulted in the decreased protein expression of EMT markers, such as MMP2 in PAAD cell lines (Figure 2h).

In conclusion, the results suggest that LRRC8A plays a significant role in the tumorigenesis and progression of PAAD.

### 3.4. Co-Expression, Gene Network, and Enrichment Analysis of LRRC8A in PAAD

The above-mentioned findings suggest that LRRC8A is a promising prognostic biomarker of PAAD. Therefore, it would be crucial to explore the potential relevance and underlying mechanisms of LRRC8A expression in patients with PAAD. Next, the LRRC8A co-expression and its network in patients with PAAD were explored, and the potential co-expression of LRRC8A genes was analyzed using eBioPortal. In PAAD database (TCGA, Firehose Legacy), the top 50 positively correlated genes of LRRC8A were imported to OmicShare to analyze the GO enrichment and KEGG pathways (Figure 3). Figure 3a shows that biological processes, including skin development, hemidesmosome assembly, epidermis development, tissue development, keratinocyte differentiation, epidermal cell differentiation, and cell junction, were associated with the tumorigenesis and progression of PAAD. Cell adhesion molecule binding, cadherin binding, protein binding involved in cell–cell adhesion, protein binding involved in cell adhesion, cadherin binding involved in cell–cell adhesion, and death receptor binding were the most highly enriched molecular functions, whereas adherence junction, anchoring junction, cell junction, membrane-bound vesicles, vesicles, extracellular region, and cell–substrate junction were the most highly enriched cellular components. Based on the KEGG pathway analysis, the phosphoinositide 3-kinase (PI3K)-Akt signaling pathway, focal adhesion, arrhythmogenic right ventricular cardiomyopathy, extracellular membrane–receptor interaction, and mitogen-activated protein kinase signaling pathway were significantly associated with the tumorigenesis and progression of PAAD (Figure 3b). Moreover, given that LRRC8A co-localizes with LRRC8B–E in cells, it is necessary to analyze the PPI network of LRRC8A–E (Figure 3c), the gene–gene interaction network of LRRC8A–E, and the top 50 positively correlated genes of LRRC8A (Figure 3d).

### 3.5. Correlation of LRRC8A with Immune Infiltration

Interestingly, the co-expression analysis of LRRC8A in PAAD samples showed that purinergic receptor P2Y2 (P2RY2) and ephrin-A receptor 2 (EPHA2) showed a positive correlation with LRRC8A in most database of PAAD, including those of TCGA (PanCancer Atlas and Firehose Legacy), QCMG (Nature 2016) [26], and TIMER (Figure 4). P2RY2 was reported to enhance cancer cell glycolysis by PI3K/AKT-mammalian target of rapamycin (mTOR) signaling, resulting in PAAD progression [27]. However, EPHA2 was reportedly involved in multiple immunity regulation pathways of PAAD [28,29]. The mRNA expression level of EPHA2 decreased considerably when LRRC8A was knocked down in AsPC-1^GEM-R^ and BxPC-3 cells (Figure 5a). Recent studies showed that LRRC8A has a novel role in immune regulation [20,30]; therefore, whether its expression was correlated with immune infiltration levels using TIMER was investigated (Figure 5b). Results showed that LRRC8A expression was significantly correlated with CD8^+^ T cells (*p* = 1.83 × 10^−3^), neutrophils (*p* = 1.58 × 10^−2^), and dendritic cells (*p* = 7.88 × 10^−3^). Similarly, a correlation analysis of LRRC8A and CD8^+^ T cells in PAAD (179 samples) using the TISIDB database also showed that LRRC8A expression was significantly correlated with CD8^+^ T cells (ρ = 0.181, *p* = 0.0155) (Figure 5c). Further immunohistochemical results of pathological sections obtained from patients with PAAD showed that LRRC8A expression was correlated with CD8 expression (Figure 5d–f), with a correlation coefficient (R^2^) of 0.83. These findings suggest that LRRC8A affects the patient clinical outcomes by interacting with immune infiltration, especially CD8^+^ T cells in PAAD. It should be noted that LRRC8C was significantly correlated with B cells (*p* = 1.67 × 10^−7^), CD8^+^ T cells (*p* = 2.22 × 10^−16^), CD4^+^ T cells (*p* = 8.02 × 10^−5^), macrophages (*p* = 7.21 × 10^−26^), neutrophils (*p* = 5.30 × 10^−20^), and dendritic cells (*p* = 3.14 × 10^−27^) (Appendix A).

## 4. Discussion

A limitless replicative potential, sustained angiogenesis, evasion of apoptosis, self-sufficiency in growth signals, and insensitivity to anti-growth signals, tissue invasion, and metastasis are classical hallmarks of cancer cells [12,13]. VRAC, one of the most important chloride channels located at the cell membrane and constituted by LRRC8A and four other homologous family members LRRC8B–E, has been implicated in each hallmark of cancer [31]. By using non-specific pharmacological VRAC blockers such as DIDS, DCPIB, and NPPB, VRAC was reported to be involved in the cell cycle regulation in human ovarian cancer, cervical cancer, nasopharyngeal carcinoma, small-cell lung cancer and glioblastoma [14]. Moreover, VRAC activity has also been implicated in the migration and multi-drug resistance in multiple cancers. In particular, VRAC current was record in pancreatic duct adenocarcinoma cell lines [32]. Hence, VRAC represents an emerging therapeutic target of PAAD. As the component of VRAC, LRRC8A–E was reported to be related to cancer cell proliferation, metastasis, and multidrug resistance in some studies, suggesting that LRRC8A–E may be the specific molecular regulated tumor progression [14]. However, the biological function and prognostic value of LRRC8A–E in PAAD have not been well-characterized.

In this study, the expression of LRRC8A–E in pan-cancer was explored using independent datasets in ONCOMINE and TCGA data in GEPIA. LRRC8A expression was higher in PAAD tissues than that in normal tissues, and it increased as PAAD progressed. Cell experiments in vitro also confirmed the high expression of LRRC8A in PAAD cells. The immunohistochemical results of pathological sections obtained from patients with PAAD also demonstrated that the high expression of LRRC8A was associated with a higher tumor grade of PAAD. Moreover, a high LRRC8A expression in patients with PAAD was significantly associated with worse clinical outcomes based on the results obtained using GEPIA and Kaplan–Meier plotter. Fortunately, two research groups almost simultaneously reported that LRRC8 heteromers were essential components of VRAC, and LRRC8A is an indispensable VRAC subunit. Therefore, the critical role and molecular mechanisms of LRRC8A in cancer management need further clarification [33,34].

Because proliferation, migration, invasion, and drug resistance are the main hallmarks of cancer cells, whether LRRC8A knockdown can inhibit the migration, invasion, and drug resistance of PAAD cells in vitro was further investigated. The wound healing assay, transwell invasion assay, and drug sensitivity assay demonstrated the inhibitory effects of LRRC8A knockdown, suggesting its potential as a therapeutic target in PAAD. Then, the functions of LRRC8A were investigated using the co-expression module of eBioPortal and GO enrichment analysis. The KEGG pathway enrichment analysis was conducted to explore the probable mechanisms in PAAD. The neighboring genes of LRRC8A were mainly enriched in the PI3K-AKT pathway and focal adhesion pathway. A previous study found that PI3K/AKT was a downstream signal responsible for the regulatory effects of LRRC8A in cancer cell proliferation [18]. The LRRC8A-PI3K/AKT-mTOR pathways can also regulate the migration of esophageal squamous cell carcinoma [17].

More importantly, EPHA2 showed a positive correlation with LRRC8A in most studies of PAAD, and its knockdown resulted in a decrease in EPHA2 expression, which was demonstrated to be involved in anti-tumor immunity by regulating prostaglandin-endoperoxide synthase. Moreover, LRRC8A is essential for T cell development [35], and it has been recently reported to play an integral role in tumor immunotherapy [20,30]. Heteromeric VRAC channels composed of LRRC8A and either LRRC8C or LRRC8E can transport cGAMP into bystander cells to activate innate immune cGAS–cGAMP–STING pathway in vascular epithelial cells and murine macrophages. The immune infiltration findings in TIMER also suggested that LRRC8A affects patient clinical outcomes by interacting with immune infiltration in PAAD. However, immune infiltration analysis also showed that LRRC8C was significantly correlated with the infiltration levels of various immune cells. It was speculated that heteromeric LRRC8A/C channels mediate cGAMP transport and p53 signaling in T cells [36].

## 5. Conclusions and Prospects

In conclusion, LRRC8A can affect the prognosis of PAAD and was correlated to cell proliferation, cell migration, drug resistance, and immune infiltration. Specifically, functional analysis indicated that LRRC8A influences the progression and prognosis of PAAD via the regulation of CD8^+^ T cells immune infiltration. We hope that our results provide novel insights into assisting in the designing of new therapeutic drugs for PAAD.

## Figures and Tables

**Figure 1 cancers-14-05526-f001:**
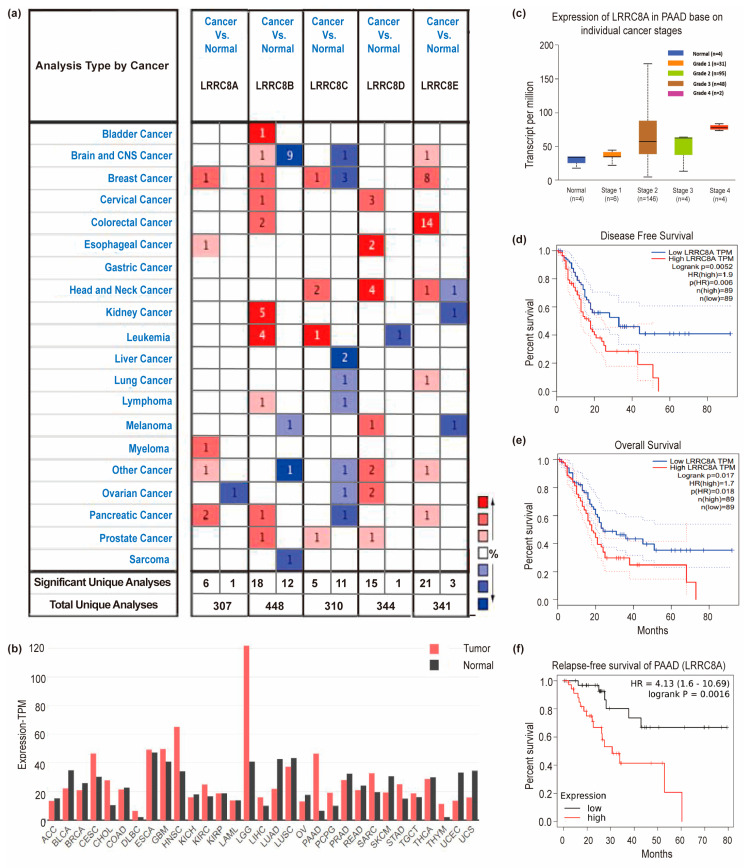
Bioinformatics analyses of LRRC8A in PAAD. (**a**) Pan-cancer analysis of LRRC8A–E expression by ONCOMINE; (**b**) the expression of LRRC8A in PAAD was seven-fold higher than that in normal by GEPIA, Tumor (n = 179), Normal (n = 171); (**c**) LRRC8A expression in PAAD based on individual cancer stages; (**d**,**e**) the DFS and OS of patients with PAAD with a high LRRC8A expression were significantly worse than those with a low LRRC8A expression; (**f**) the RFS of patients with PAAD with a high LRRC8A expression were significantly worse than those with a low LRRC8A expression.

**Figure 2 cancers-14-05526-f002:**
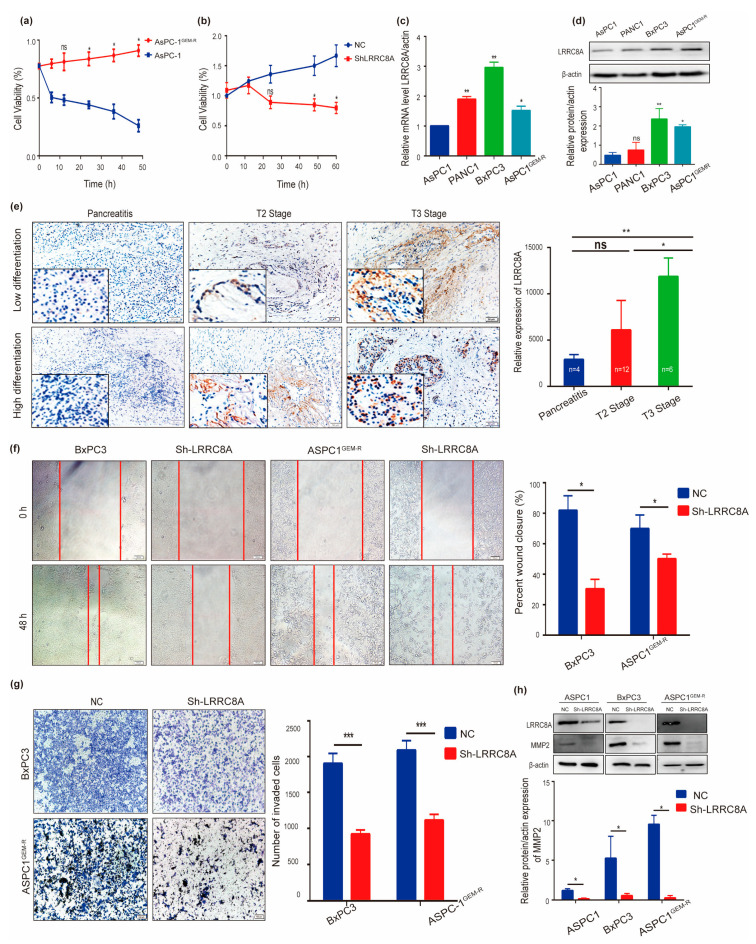
Roles of LRRC8A in PAAD in vitro. (**a**) Cell viability of AsPC-1 and AsPC-1^GEMR^ cells on GEM; (**b**) cell viability of AsPC-1^GEMR^ cells on GEM after pretreated with shLRRC8A or scramble lentivirus (NC); (**c**) mRNA expression of LRRC8A in PAAD cell lines; (**d**) LRRC8A protein expression in PAAD cell lines; (**e**) representative immunohistochemical results of LRRC8A in patients with PAAD and quantitative analysis results; (**f**) wound healing assay using BxPC-3 cells and AsPC-1^GEM-R^ cells with LRRC8A knockdown and quantitative analysis results; (**g**) transwell experiments using AsPC-1 and BxPC-3 cells, and quantitative analysis results; (**h**) expression of LRRC8A and MMP2 after AsPC-1, AsPC-1^GEM-R^ and BxPC-3 cells were treated by shLRRC8A. Error bars indicate SE. * *p* < 0.05, ** *p* < 0.01 and *** *p* < 0.001 by Student’s t-tests. The whole western blots were shown in Appendix A. ns: not significant.

**Figure 3 cancers-14-05526-f003:**
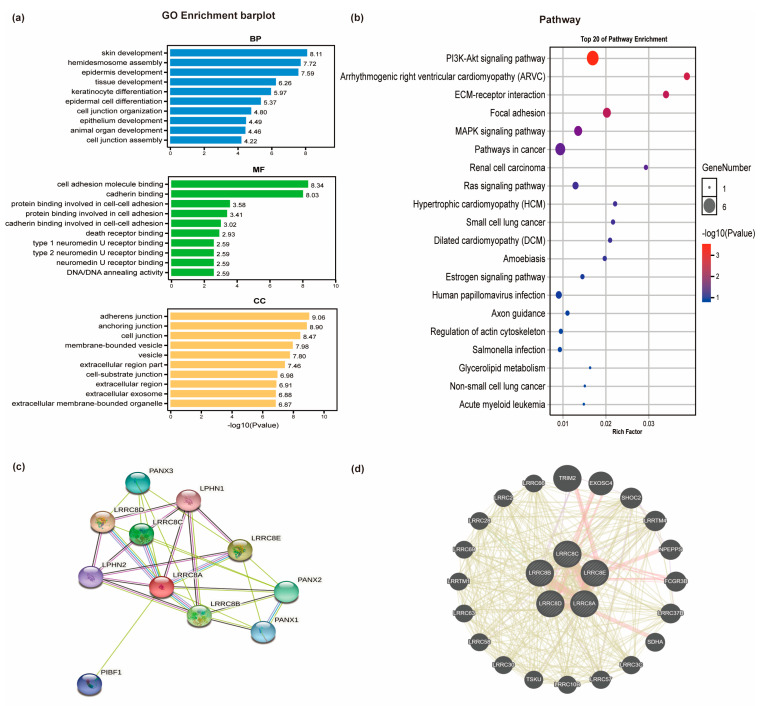
Function analysis of LRRC8A in PAAD. (**a**) GO enrichment analysis (biological process, molecular function, and cellular component) of the top 50 positively correlated genes of LRRC8A; (**b**) KEGG enrichment analysis of the top 50 positively correlated genes of LRRC8A; (**c**) protein interaction network of LRRC8A–E; (**d**) gene–gene interaction network of LRRC8A–E and the top 50 positively correlated genes of LRRC8A.

**Figure 4 cancers-14-05526-f004:**
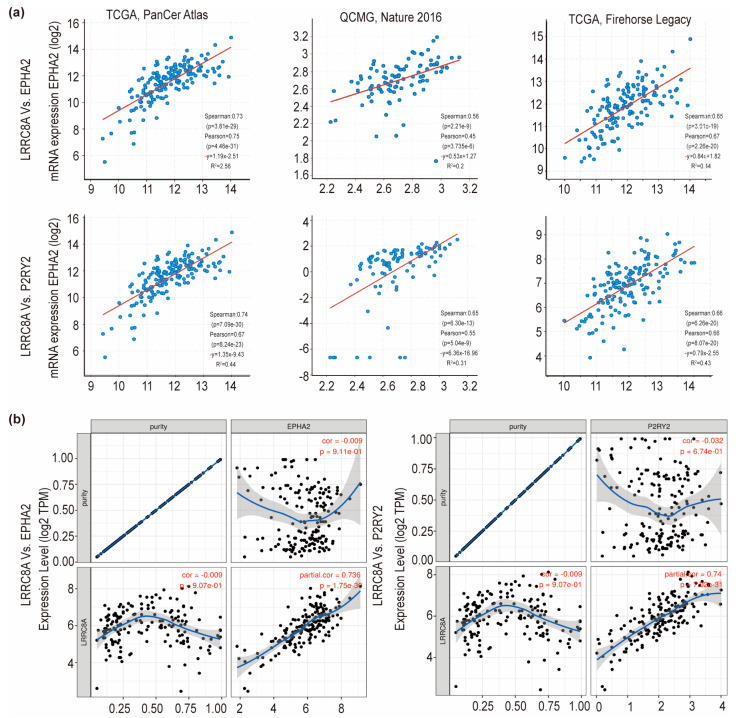
Correlation analysis of LRRC8A with P2RY2 and EPHA2 in PAAD. (**a**) LRRC8A showed a positive correlation with P2RY2 and EPHA2 in cBioPortal; (**b**) LRRC8A showed a positive correlation with P2RY2 and EPHA2 in TIMER.

**Figure 5 cancers-14-05526-f005:**
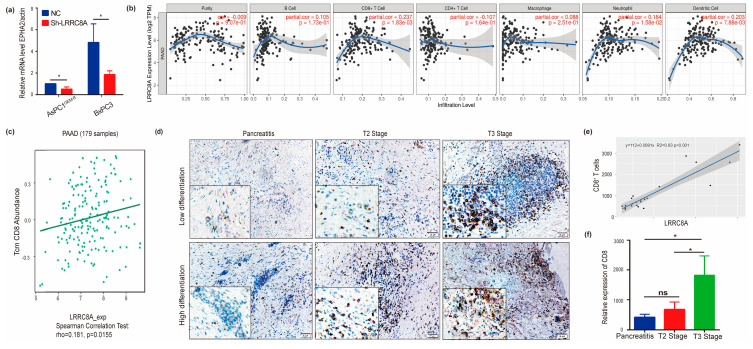
Immune infiltration analysis of LRRC8A in PAAD. (**a**) EPHA2 expression decreased in BxPC3 (*p ** < 0.05) and AsPC1^GEM-R^ cells (*p ** < 0.05) after LRRC8A knockdown; (**b**) immune infiltration of LRRC8A in PAAD; (**c**) correlation analysis of LRRC8A and CD8^+^ T cells; (**d**) representative immunohistochemical results of CD8 in patients with PAAD; (**e**) correlation analysis of LRRC8A and EPHA2 in patients with PAAD; (**f**) quantitative analysis results of Figure 5d. ns: not significant.

**Table 1 cancers-14-05526-t001:** Primer sequences for Real-time PCR.

Gene Name	Sequences (5′ to 3′)
LRRC8A FORWARD	GGAGAGCAGCTACAGCGACA
LRRC8A REVERSE	TCACTCACCTCCGACAGGAA
EPHA2 FORWARD	AGAAGCGCCTGTTCACCAAG
EPHA2 REVERSE	GCTCCTCCACGTTCAGCTTC
ACTIN FORWARD	CATGTACGTTGCTATCCAGGC
ACTIN REVERSE	CTCCTTAATGTCACGCACGAT

## Data Availability

Authors declared that all and the other data supporting the findings of this study are available within the paper. The raw data that support the findings of this study are available from the corresponding author upon reasonable request.

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
