# Peer review of "LRRC8A Is a Promising Prognostic Biomarker and Therapeutic Target for Pancreatic Adenocarcinoma"

_cancers, 2022, doi:10.3390/cancers14225526_

Round 1
Reviewer 1 Report (Previous Reviewer 3)
It is an interesting manuscript and was improved with the revision
Author Response
Thank you very much for your work and suggestion on our manuscript entitled “LRRC8A is a promising prognostic biomarker and therapeutic target in pancreatic adenocarcinoma" (Manuscript Number: cancers-1979476).
Thanks again for your comments.
Reviewer 2 Report (Previous Reviewer 2)
I am still not really convinced by the authors response but also by their scientific rigour. For instance, in their letter to reviewers, they wrote "Figure 1d and 1e" instead of "Figure 2d and 2e". In the original manuscript, the supplementary Figure 1 a was in the main text. In Figure 2D, the authors mentionned RT-qPCR data that was not even in the Figure. A lot of small "mistakes" that make me feel that the authors lack a certain scientific accuracy and rigour in writing the mansucript. I am concerned that they have applied this " low level of rigour" in obtaining their scientific results. I might be wrong. Because of this weird feeling, I will let to the Editor-in-chief the decision to accept or reject this manuscript.
Author Response
Octorber 13, 2022
Journal of Cancers
Dear Editors and reviewers:
Thank you for your work and suggestion on our manuscript entitled “LRRC8A is a promising prognostic biomarker and therapeutic target in pancreatic adenocarcinoma" (Manuscript Number: cancers-1979476). Those comments are indeed valuable and very helpful for improving our paper further. We have made corrections according to the comments carefully and which we hope you to consider for publication in the Journal of Cancers. A point-by-point response is described below.
Reviewer #1:
- I am still not really convinced by the authors response but also by their scientific rigour. For instance, in their letter to reviewers, they wrote "Figure 1d and 1e" instead of "Figure 2d and 2e". In the original manuscript, the supplementary Figure 1 a was in the main text. In Figure 2D, the authors mentionned RT-qPCR data that was not even in the Figure. A lot of small "mistakes" that make me feel that the authors lack a certain scientific accuracy and rigour in writing the mansucript. I am concerned that they have applied this " low level of rigour" in obtaining their scientific results. I might be wrong. Because of this weird feeling, I will let to the Editor-in-chief the decision to accept or reject this manuscript.
Response: Thank you for your careful work and good suggestion. I feel so sorry for my carelessness and mistakes in the response letter to you. I have carefully checked the manuscript again and deleted the supplementary Figure 1a in the main text. The supplementary figures can be found in the compressed package “figures” in submission files. Besides, I adjusted the format of the reference list, citations style and uploaded the revised manuscript and another tracked version for you. The RT-qPCR data of PAAD cell lines was shown in Figure 2c in the manuscript and Figure 2d presented the Western blot results of LRRC8A in PAAD cell lines and corresponding quantitative results.
The relative description was supplied in the second paragraph in 3.3:
“To evaluate the potential role of LRRC8A in PAAD, qRT-PCR and western blotting were used to detect the expression of LRRC8A in PAAD cell lines AsPC-1, PaNC-1, and BxPC-3. The sequence of cell migration and invasion capacity, angiogenic potential, and tumorigenicity was reportedly BxPC-3 > AsPC-1 > PaNC-1 [25]. A similar mRNA and protein expression gradient of LRRC8A in PAAD cell lines was also observed in Figure 2c and Figure 2d, respectively.”
Thank you for your consideration. I look forward to hearing from you.
Sincerely,
Changhong Shi, M.D., Ph.D.
Professor of Division of Cancer Biology, Laboratory Animal Center
The Fourth Military Medical University
E-mail: changhong@fmmu.edu.cn; Phone: +86-29-84774787

This manuscript is a resubmission of an earlier submission. The following is a list of the peer review reports and author responses from that submission.
Round 1
Reviewer 1 Report
I have carefully reviewed the manuscript entitled: "LRRC8A is a promising prognostic biomarker and therapeutic target for pancreatic adenocarcinoma " by Xu et al. In this work, Xu et al. did bioinformatics analysis based on several large public databases and identified LRRC8A as a possible potent therapeutic target and prognostic biomarker for PDAC based on cell experiments. This paper is clearly written and well organized. The introduction and background are reasonable, and the figures are comprehensive. However, I have a few minor concerns that could be addressed to improve the quality of this study.
Minor comments:
- What is the effect of LRRC8A knockdown on the protein expression levels of EMT markers in Pancreatic ductal carcinoma cell lines?
Author Response
Reviewer #1:
- What is the effect of LRRC8A knockdown on the protein expression levels of EMT markers in Pancreatic ductal carcinoma cell lines?
Response: Thank you for your careful work and good suggestion. We detected the expression of EMT-related markers, MMP2 using Western blot. The relative result was added in Figure 2h, and it demonstrated that the expression of MMP2 decreased significantly after LRRC8A was knocked down using shLRRC8A in AsPC-1, BxPC3 and AsPC-1GEMR cells, indicating LRRC8A may regulate cell migration and invasion by MMP2-related signal pathways.
The relative description was supplemented in revision as following:
“
Reviewer 2 Report
Dear Editor,
I read with a great interest the manuscript by Xu et al. entitled “LRRC8A is a promising prognostic biomarker and therapeutic target for pancreatic cancer”. This study mainly uses transcriptomic public data sets to demonstrate the putative role of LRRC8A as biological marker for pancreatic cancer. The manuscript s relatively well written but the description of the results is often confusing especially the in vitro data (see comments below). This manuscript must be largely modified. I regret but I have no other options to advice a rejection of this manuscript for publication in Cancers
In Figure 1, the authors LRRC8 tend to demonstrate that LRRC8A is overexpressed in PDAC samples based on ONCOMINE transcriptomic data bases. They further compare LRRC8 expression in different PDAC grades using GEPIA and UALCAN public data sets in Fig 1B. They claim an overexpression of LRRC8A in PDAC samples compared to normal tissues and an increase dependent in the stage of the tumor. However, it is always dangerous to compare expression level of any mRNA to normal tissues that are phenotypically different (in the case of PDAC, “normal tissues” are mostly composed of acini). What is the origin of normal tissues ? In addition, the number of samples in normal tissues, stage 1 stage 3 and stage 4 are extremely small ( n=4 vs stage 2 samples (n=178). The conclusions are therefore overstated with such a small number of samples in the different stages.
Minor points: The figure legends 1 b and 1 f contains mostly “results” content. Two decimals for the fold induction (46.52 vs 6.59) is not necessary especially for this type of analysis.
Minor point: why the supplementary Figure 1 is the main manuscript?
Major point: The identification of the role of LRRC8 in PDAC in vitro is very poor. In Fig 2A, the dose of gemcitabine to generate AsPC-1 resistant cell line is not mentioned. Moreover, the authors use the DCPIB, a VRAC inhibitor, that they qualify in the Materials and Methods section as a “a kind of specific VRAC blocker”. What does I mean ? The Figure 2C show a dramatic reduction of LRRC8 mRNA reduction in response to DCPIB treatment. I would have like to see a Western blot assay to confirm this quasi-total reduction. In figure 2D, they authors write “to evaluate the potential role of LRRC8A in PAAD, qRT-PCR and western blotting were used to detect the expression of LRRC8A in PAAD cell lines AsPCT-1, PANC-1 and BxPC-3”. I do not see the data of qRT-PCR. In the text corresponding to the Figure 2F, the authors describe a wound healing assay and a transwell invasion assay in BxPC-3. They do not even mention that they use a ShRNA against LRRC8. Moreover, AsPC-1 is mentioned on the Figure that is not even described in the result description.
Minor point: in the paragraph 3.3, PANC-1 instead of PaNC1.
Author Response
Reviewer #2:
- In Figure 1, the authors LRRC8 tend to demonstrate that LRRC8A is overexpressed in PDAC samples based on ONCOMINE transcriptomic data bases. They further compare LRRC8 expression in different PDAC grades using GEPIA and UALCAN public data sets in Fig 1B. They claim an overexpression of LRRC8A in PDAC samples compared to normal tissues and an increase dependent in the stage of the tumor. However, it is always dangerous to compare expression level of any mRNA to normal tissues that are phenotypically different (in the case of PDAC, “normal tissues” are mostly composed of acini). What is the origin of normal tissues? In addition, the number of samples in normal tissues, stage 1 stage 3 and stage 4 are extremely small (n=4 vs stage 2 samples (n=178). The conclusions are therefore overstated with such a small number of samples in the different stages.
Response: Thanks for your comments and it’s of great benefit to our research. The analysis using mRNA data of PAAD and normal tissue indeed cannot fully reflect the difference between normal pancreas and pancreatic cancer. Because the public dataset of PAAD that we can obtained are mostly transcription group public dataset and protein expression level were usually conducted using their own samples in lots of researches on pancreatic cancer (PMID: 32737864). In order to clarify the effects of LRRC8A on PAAD, the protein expression of LRRC8A in multiple PAAD cell lines and patients with or without PAAD was also detected using Western blot and IHC in our research, such as Figure 1d and 1e.
The relative description was supplied as following in the discussion of revision:
“LRRC8A expression was higher in PAAD tissues than that in normal tissues, and it increased as PAAD progressed. Cell experiments in vitro also confirmed the high expression of LRRC8A in PAAD cells. The immunohistochemical results of pathological sections obtained from patients with PAAD also demonstrated that the high expression of LRRC8A was associated with a higher tumor grade of PAAD.”
As for the origin of normal tissues, normal adjacent tissues or the entire normal tissues, such as atrophic pancreas with a single focus of low-grade PanIN or normal pancreas with atrophy were set as control in pan-cancer analysis, which can be found in supplement file in TCGA. Besides, the number of samples in stage 1 stage 3 and stage 4 is indeed too small, but the number of samples in different stages depends on the source of samples. In the research of PMID: 34873463 and PMID: 34017995, the samples were also divided according to the stages and the number of samples in different stages was small. So, further tests on human samples will be important in the following study.
- The figure legends 1 b and 1 f contains mostly “results” content. Two decimals for the fold induction (46.52 vs 6.59) is not necessary especially for this type of analysis.
Response: This is very valuable suggestions. I have deleted the “results” content in legend of Figure 1.
- why the supplementary Figure 1 is the main manuscript?
Response: Thanks for your careful work. I moved the supplementary Figures at the end of the manuscript.
- The identification of the role of LRRC8 in PDAC in vitro is very poor. In Fig 2A, the dose of gemcitabine to generate AsPC-1 resistant cell line is not mentioned.
Response: I have revised these parts carefully. The gemcitabine-resistant AsPC-1 cell line was generated by treating AsPC-1 cell with medium containing 0.8 μg/mL gemcitabine in the beginning, and then changed medium without gemcitabine and passaged the cells, then treating cells with higher gemcitabine concentration, repeated the above process until cells can grow in the medium containing 18 μg/mL gemcitabine for more than 4 days.
These contents are supplemented in the 2.6. Cell lines part of revision.
“PAAD cell lines PANC1 (CRL-1469™), AsPC-1 (CRL-1682™), and BxPC-3 (CRL-1687) were obtained from the American Type Culture Collection (USA), AsPC-1GEM-R cell line (gemcitabine-resistant AsPC-1) was established in our laboratory, which can grow in the medium containing 18 μg/mL gemcitabine for more than 4 days.”
- Moreover, the authors use the DCPIB, a VRAC inhibitor, that they qualify in the Materials and Methods section as a “a kind of specific VRAC blocker”. What does I mean? The Figure 2C show a dramatic reduction of LRRC8 mRNA reduction in response to DCPIB treatment. I would have like to see a Western blot assay to confirm this quasi-total reduction.
Response: Thanks for your careful work. I have uploaded the mRNA and protein expression of LRRC8A in AsPC-1GEMR cells after treated with DCPIB and gemcitabine in supplementary Figure 1d. Moreover, in order to illustrate the effect of LRRC8A, I added Cell viability of AsPC-1GEMR cells on GEM after pretreated with shLRRC8A in Figure 1b, and the knockdown effect of shLRRC8A in PAAD cell lines was showed in Figure 2h and supplementary Figure 1e.
These contents can be found in the 3.3. Identification of the role of LRRC8A in PAAD in vitro.
“The CCK-8 assay and qRT-PCR results revealed that the knockdown of LRRC8A using DCPIBshRNA or DCPIB, the specific VRAC inhibitor can inhibit the proliferation of PAAD cells and increase their sensitivity to gemcitabine (Figure 2B2b, 2C Supplementary Figure 1d). Besides, qRT-PCR and Western blot results showed LRRC8A decreased after AsPC-1GEMR cells were pretreated with DCPIB (Supplementary Figure 1d) and the knockdown effect of shLRRC8A in PAAD cell lines was showed in Supplementary Figure 1e.”
- In figure 2D, they authors write “to evaluate the potential role of LRRC8A in PAAD, qRT-PCR and western blotting were used to detect the expression of LRRC8A in PAAD cell lines AsPC-1, PANC-1 and BxPC-3”. I do not see the data of qRT-PCR.
Response: Thanks for your professional suggestion. It is my mistakes to miss the result of LRRC8A mRNA expression in PAAD cell lines AsPC-1, PANC-1 and BxPC-3. I added the relative results in Figure 2c of revision.
- In the text corresponding to the Figure 2F, the authors describe a wound healing assay and a transwell invasion assay in BxPC-3. They do not even mention that they use a ShRNA against LRRC8. Moreover, AsPC-1 is mentioned on the Figure that is not even described in the result description.
Response: Thanks for your careful work. Because the expression of LRRC8A is relative high in BxPC-3 and AsPC-1GEMR cells, so we did wound healing assay and transwell invasion assay in these cells as shown in Figure 2f and 2g. The result description was also changed as shown in the revised manuscript.
“To investigate whether LRRC8A affects the migration and invasion abilities of PAAD cells, a wound healing assay with or without LRRC8A knockdown was conducted using AsPC-1GEMR and BxPC-3 cells as showed in Figure 2f, and a transwell invasion assay using AsPC-1 and BxPC-3 cells with or without LRRC8A knockdown were performed in Figure 2g. Results showed a substantial inhibitory effect of LRRC8A knockdown on PAAD cell migration and invasion (Figure 2f, 2g). Further investigation showed LRRC8A knockdown resulted in the decreased protein expression of EMT markers, such as MMP2 in PAAD cell lines (Figure 2h).”
Reviewer 3 Report
A number of GWAS have been performed for PDAC, and shows the genetic backround of the disease. The authors did not discuss them and it is unlcear how the author conclude to study LRRC8A
Additionally tests on human samples should be important and the clinical impact of the findings should be better discussed
Author Response
- A number of GWAS have been performed for PDAC, and shows the genetic backround of the disease. The authors did not discuss them and it is unlcear how the author conclude to study LRRC8A
Additionally tests on human samples should be important and the clinical impact of the findings should be better discussed
Response: Thanks for your valuable comments and suggestions. The reason why we decided to study the function of LRRC8A in PAAD can be summarized into 3 aspects. Firstly, given the purported significance of VRAC in tumorigenesis and tumor progression, together with its positive expression in PAAD cells, VRAC was regarded as a potential therapeutic target of PAAD. Secondly, VRAC was assembled by LRRC8A and up to four other homologous family members (LRRC8B-E) and LRRC8A was the main component involved in regulation of tumorigenesis and tumor progression in various cancers, indicating LRRC8A may be the molecular therapeutic target of PAAD. Thirdly, by using a series of bioinformatics analyses and cell experiments, LRRC8A was indeed implicated in cell migration, drug sensitivity and immune modulation in PAAD, which can provide a basis for development of molecular therapeutic target of PAAD, especially immune related target. Besides, the discussion why we focused the function of VRAC and LRRC8A in PAAD can be found in the first and second paragraph of discussion.
Furthermore, we also did some tests on human samples, such as the protein expression of LRRC8A and CD8 was tested on pathological section of patients with PAAD, as showed in Figure 2e and Figure 5d. To be honest, the number of human samples is indeed small and a further cohort study on PAAD and LRRC8A will be conducted later.
The relative description was supplemented in the first and second paragraph of discussion.
“VRAC, one of the most important chloride channel located at the cell membrane and constituted by LRRC8A and four other homologous family members LRRC8B–E, has been implicated in each hallmark of cancer [31]. By using non-specific pharmacological VRAC blockers such as DIDS, DCPIB, and NPPB, VRAC was reported to be involved in the cell cycle regulation in human ovarian cancer, cervical cancer, nasopharyngeal carcinoma, small-cell lung cancer and glioblastoma [14]. Besides, VRAC activity has also been implicated in the migration and multi-drug resistance in multiple cancers. In particular, VRAC current was record in pancreatic duct adenocarcinoma cell lines [32]. Hence, VRAC represents an emerging therapeutic target of PAAD. As the component of VRAC, LRRC8A–E was reported to be related to cancer cell proliferation, metastasis, and multidrug resistance in some studies, suggesting that LRRC8A–E may be the specific molecular regulated tumor progression [14]. However, the biological function and prognostic value of LRRC8A–E in PAAD have not been well-characterized.”
“In this study, the expression of LRRC8A–E in pan-cancer was explored using independent datasets in ONCOMINE and TCGA data in GEPIA. LRRC8A expression was higher in PAAD tissues than that in normal tissues, and it increased as PAAD progressed. Cell experiments in vitro also confirmed the high expression of LRRC8A in PAAD cells. The immunohistochemical results of pathological sections obtained from patients with PAAD also demonstrated that the high expression of LRRC8A was associated with a higher tumor grade of PAAD. Moreover, a high LRRC8A expression in patients with PAAD was significantly associated with worse clinical outcomes based on the results obtained using GEPIA and Kaplan–Meier plotter. Fortunately, two research groups almost simultaneously reported that LRRC8 heteromers were essential components of VRAC, and LRRC8A is an indispensable VRAC subunit. Therefore, the critical role and molecular mechanisms of LRRC8A in cancer management need further clarification [33, 34].”